# Effects of Aluminium Oxide Content on the Regenerated Magnesia-Calcium Bricks for Cement Rotary Kiln

Gui-Bo Qiu [1,2,3,4], Yi-Dang Hao [1,2,3,5], Jia Hou [1,2,3,4], Hui-Gang Wang [1,3,4,5], Xuan-Hao Zhang [1,2,3,4], Ben Peng [1,5,*] and Mei Zhang [3,4,6,*]

[1]  Central Research Institute of Building and Construction Co., Ltd., MCC Group, Beijing 100088, China; qiuguibo@cribc.com (G.-B.Q.); haoyidang@cribc.com (Y.-D.H.); houjia@cribc.com (J.H.); wanghuigang0822@126.com (H.-G.W.); zhangxuanhao0308@163.com (X.-H.Z.)
[2]  Zhanjiang Environmental Protection Operation Management Co., Ltd., MCC Group, Zhanjiang 524000, China
[3]  Guangdong Iron and Steel Smelting Resource Comprehensive Utilization Engineering Technology Research Center, Zhanjiang 524000, China
[4]  Zhanjiang Key Laboratory of Environmental Protection for Metallurgical Industry, Zhanjiang 524000, China
[5]  Energy Conservation and Environment Protection Co., Ltd., MCC Group, Beijing 100088, China
[6]  School of Metallurgical and Ecological Engineering, University of Science and Technology Beijing, Beijing 100083, China
*  Correspondence: pengben@cribc.com (B.P.); zhangmei@ustb.edu.cn (M.Z.)

**Abstract:** Regenerated magnesia-calcium brick samples with different aluminium oxide ($Al_2O_3$) contents were prepared using spent magnesia-calcium bricks and fused magnesia as the main raw materials and $Al_2O_3$ powders as the additive. The phase compositions, microstructures, room temperature, hot flexural strength, and kiln coating adherence of the regenerated samples were investigated. This indicates that the $Al_2O_3$ content increased, mainly resulting in the content of tetracalcium aluminoferrite ($C_4AF$) and tricalcium aluminate ($C_3A$) increasing in the regenerated samples. The bulk density, room temperature flexural strength, and kiln coating adherence all increased, whereas the hot flexural strength and corrosion resistance to cement clinker both deteriorated with an increase in the $Al_2O_3$ content. This was because, on the one hand, the low melting point phases of $C_4AF$ and $C_3A$ improved the sinterability of the regenerated samples during the burning stage, and on the other hand, they melted or existed in the liquid phase at the experimental temperature, which degraded the hot flexural strength and corrosion resistance but enhanced the kiln coating adherence as the wettability of the liquid phase. The content of $Al_2O_3$ in the regenerated magnesia-calcium brick should not be higher than 1.1 wt.%, considering its comprehensive performance for cement rotary kiln.

**Keywords:** regenerated magnesia-calcium bricks; aluminium oxide; microstructures; flexural strength; kiln coating adherence

## 1. Introduction

Currently, magnesite-chrome bricks are widely used in the cement rotary kiln in China [1]. This has led to serious environmental problems because Cr(III) is easily oxidized into toxic Cr(VI) in magnesite-chrome bricks under natural conditions [2], and spent magnesite-chrome bricks have not been reasonably handled. Therefore, it is urgently needed to develop free chrome refractory bricks as substitutes for magnesite-chrome bricks in cement rotary kiln [3–5].

Magnesia-calcium bricks have been widely used as lining materials in ferrous metallurgy to produce clean steel such as in refining furnaces [6] because of their excellent performance [7]. However, there are amounts of spent magnesia-calcium bricks not being reused effectively [8] because impurity elements such as Si, Fe, and Al from molten steel and slag remain in spent magnesia-calcium bricks, and these impurities can damage the

usability of regenerated products [9–11]. These spent magnesia-calcium bricks have not been reused, as they not only occupy large amounts of land and cause environmental harm but also cause a waste of resources such as magnesia and calcium oxide in the spent bricks. At present, there are very few studies on the utilization of spent magnesia-calcium bricks [8,12].

Magnesia-calcium bricks are also regarded as a substitute for magnesite-chrome bricks for the cement rotary kiln [13] because they have similar operational performances, such as high-temperature resistance, corrosion resistance, and good coating adherence [14,15]. However, the application of magnesia-calcium bricks in the cement rotary kiln has been restricted since they have poor hydration resistance, which can result in serious structural damage [16]. Related studies have shown that aluminium oxide ($Al_2O_3$) can enhance the hydration resistance of magnesia-calcium bricks [17,18].

In a previous work [19,20], a kind of regenerated magnesia-calcium brick with high-hydration resistance was prepared based on spent magnesia-calcium bricks, without removing impurities from spent magnesia-calcium bricks. The maximum utilization rate of the spent magnesia-calcium brick was 67 wt.%. This not only leads to the high reutilization rate of spent magnesia-calcium bricks but also provides the possibility for the application of regenerated magnesia-calcium bricks in the cement rotary kiln to substitute for magnesite-chrome bricks due to the improved hydration resistance. Nevertheless, the content of $Al_2O_3$ impurities in different spent magnesia-calcium bricks was fluctuant in previous experiments; moreover, high-temperature performances, such as high-temperature strength and refractoriness, can be degraded because of excessive impurity content, particularly the content of $Al_2O_3$ [21,22]. Therefore, the content of $Al_2O_3$ impurity in the regenerated magnesia-calcium bricks might have a remarkable influence on the performance of the cement rotary kiln.

In this work, the effects of $Al_2O_3$ content on regenerated magnesia-calcium bricks for the cement rotary kiln were studied in detail by investigating the regenerated magnesia-calcium bricks with different mass ratios of $Al_2O_3$ additive to the main materials.

## 2. Materials and Methods

### 2.1. Raw Materials of the Regenerated Magnesia-Calcium Bricks

The raw materials of the regenerated magnesia-calcium bricks were divided into main materials and additives, of which the main raw materials were spent magnesia-calcium bricks and fused magnesia, and the additive was $Al_2O_3$ powders. The spent magnesia-calcium bricks were obtained from a refining furnace in the ferrous industry, the fused magnesia was obtained from a refractory materials plant, and the $Al_2O_3$ powder additive was a kind of analytical reagent with a particle size of less than 0.088 mm purchased from Sinopharm Chemical Reagent Co., Ltd. of China (Shanghai, China). The chemical compositions of the main raw materials were determined by measuring their powders with a particle size of less than 0.088 mm using an 1800 X-ray fluorescence spectrometer (XRF) with a test error of less than 0.05%, and the results are shown in Table 1.

**Table 1.** Compositions of the raw materials of the regenerated magnesia-calcium bricks (wt.%).

| Constituents | MgO | CaO | $SiO_2$ | $Fe_2O_3$ | $Al_2O_3$ |
|---|---|---|---|---|---|
| Spent magnesia-calcium bricks | 58.18 | 33.92 | 2.81 | 2.68 | 1.51 |
| Fused magnesia | 94.16 | 1.63 | 2.64 | 1.00 | 0.27 |
| $Al_2O_3$ powders | - | - | 0.30 | 0.03 | 98.75 |

### 2.2. Preparation of the Regenerated Magnesia-Calcium Bricks

Figure 1 shows the synthetic process of the regenerated magnesia-calcium bricks, which was an improvement of the previous work [19,20]. Spent magnesia-calcium bricks and fused magnesia were crushed and screened into powders (particle size < 0.088 mm), among which spent magnesia-calcium bricks powders were calcinated at 1173 K for 2 h and

fused magnesia powders were dried at 383 K for 2 h in order to remove the hydration factor from the main raw materials. The mixture of the regenerated magnesia-calcium brick green body was prepared by mixing spent magnesia-calcium bricks and fused magnesia powders, $Al_2O_3$ powder additive, and melted paraffin (purchased from Sinopharm Chemical Reagent Co., Ltd. of China) as the binder. The green body was prepared by pressing the mixture at 100 MPa to form a rectangle with dimensions of 60 mm × 8 mm × 8 mm subsequently. The regenerated magnesia-calcium brick samples (regenerated samples) were obtained by firing the green bodies in a silicon-molybdenum bar heating furnace, heating to 1873 K at a rate of 5 K/min, and maintaining for 2 h under an air atmosphere finally.

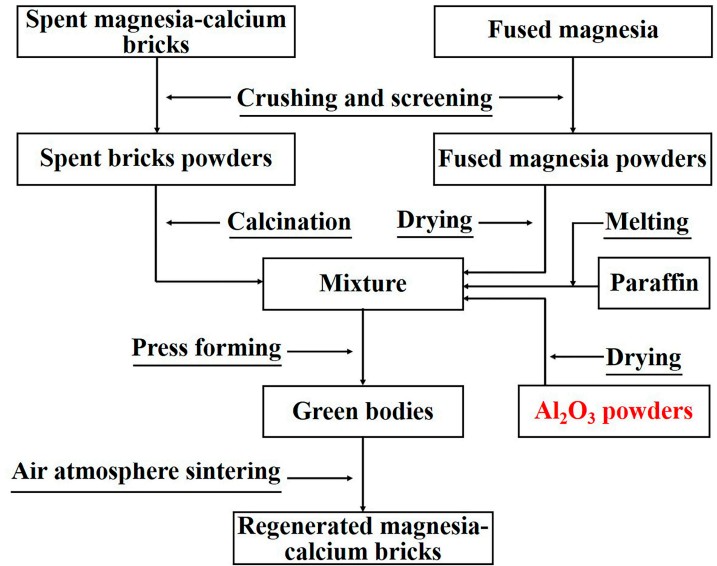

**Figure 1.** The synthetic process flow chart of regenerated magnesia-calcium bricks with $Al_2O_3$ powder addition.

In a previous work, the content of $Al_2O_3$ impurity in a regenerated sample containing 70 wt.% MgO was from 1.10 wt.% to 3.34 wt.% [13]. In order to study the effects of $Al_2O_3$ content on the regenerated samples for cement rotary kiln, in this work, three kinds of regenerated samples were prepared, among which the mass ratios of the spent magnesia-calcium bricks to the fused magnesia were all 67.15:32.85, and the mass ratios of the $Al_2O_3$ powder additive to the main materials were 0:100, 1.5:100, and 3.0:100; that is, the $Al_2O_3$ addition was 0 wt.%, 1.5 wt.%, and 3.0 wt.%, respectively. Their compositions are shown in Table 2, and their corresponding fired samples are marked as samples A0, A1.5, and A3.0. It can be seen that the $Al_2O_3$ impurity content in samples A0, A1.5, and A3.0 was from 1.10 wt.% to 3.94 wt.% (as shown in Table 2), it could cover the fluctuation range of the $Al_2O_3$ content in regenerated samples prepared in the previous work.

**Table 2.** Compositions of the regenerated magnesia-calcium bricks' green bodies (wt.%).

| Sample | MgO | CaO | SiO$_2$ | Fe$_2$O$_3$ | Al$_2$O$_3$ |
|--------|-------|-------|-------|-------|-------|
| A0 | 70.00 | 23.31 | 2.66 | 2.21 | 1.10 |
| A1.5 | 68.97 | 22.97 | 2.63 | 2.18 | 2.54 |
| A3.0 | 67.96 | 22.63 | 2.59 | 2.15 | 3.94 |

*2.3. Methods of Investigation*

2.3.1. Phase Compositions

To determine the phase compositions of the regenerated samples, their powders (<0.088 mm) were identified via MXP21VAHF X-ray powder diffractometry (XRD) analysis. The formation reactions of the impurity phases were calculated using thermodynamics.

### 2.3.2. Microstructures and Elements Distribution

To explore the microstructures of the regenerated samples, their surfaces and fractures were investigated using an MLA 250-FEI Quanta scanning electron microscope (SEM), and the element distribution of the microstructures was investigated using an energy dispersive spectrometer (EDS). In addition, the bulk density of each kind of regenerated sample was measured using the Archimedes method to characterize the compactness of the regenerated samples.

### 2.3.3. Room Temperature and Hot Flexural Strength

The three-point bending test method was used to measure the room-temperature flexural strength of the regenerated samples. The maximum force to break each regenerated sample was carried out using a WDW-10E microcomputer-controlled electronic universal testing machine.

The hot flexural strength of the regenerated samples was determined using the three-point bending test method. Each regenerated sample was placed into a CMT 5204 temperature-mechanical load coupling testing machine and heated to 1573 K at a rate of 5 K/min. The maximum force required to break each kind of regenerated sample was determined after maintaining 1573 K for 30 min.

### 2.3.4. Kiln Coating Adherence

Kiln coating adherence refers to the ability of the refractory lining to react with cement to form a protective layer during the formation of cement. Suitable kiln coating adherence can protect refractories, inhibit their further erosion, and extend their service life. Therefore, kiln coating adherence is one of the most important properties of cement rotary refractories.

Two identical regenerated samples of each batch, A0, A1.5, and A3.0, were bonded with 5 mm thick cement clinker grout that contained 90.48 wt.% cement clinker, 4.76 wt.% $K_2SO_4$, and 4.76 wt.% glycerol to make the test sample, as shown in Figure 2a, among which the cement clinker was a kind of 425 Portland cement clinker from a cement plant, and $K_2SO_4$ and glycerol were both kinds of analytical reagents. These test samples were named bonded samples BA0, BA1.5, and BA3.0, and they were placed in a silicon-molybdenum bar heating furnace, heated to 1823 K at a rate of 5 K/min, and kept at 1823 K for 3 h. After cooling to room temperature, the maximum breaking force of each test sample was determined using the three-point bending test method with a WDW-10E microcomputer-controlled electronic universal testing machine, as shown in Figure 2b. The flexural strengths of the bonded samples were determined using the three-point bending test method, which was used to characterize the kiln coating adherence of the regenerated sample; that is, the higher the flexural strengths of the bonded sample, the better the kiln coating adherence of the regenerated sample. In addition, to explore the reaction mechanism between kiln coating and regenerated samples, the microstructure and element distributions of the bonded samples after the kiln coating adherence experiment were investigated using an MLA 250-FEI Quanta SEM and an EDS.

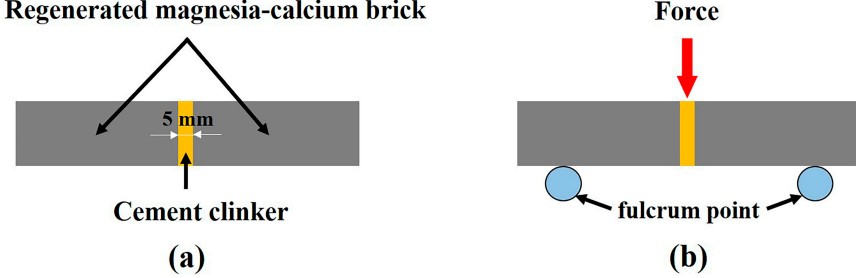

**Figure 2.** (**a**) Preparation diagram of the bonded sample for the kiln coating adherence test; (**b**) flexural strength diagram of the bonded sample for the kiln coating adherence test.

## 3. Results

### 3.1. Phase Compositions of the Regenerated Magnesia-Calcium Bricks

Figure 3a shows the XRD patterns of samples A0, A1.5, and A3.0. The results indicate that the main phase compositions of the samples A0 and A1.5 were identical to those of magnesia (MgO), free calcium oxide (f-CaO), tricalcium silicate ($Ca_3SiO_5$, $C_3S$), and tetracalcium aluminoferrite ($Ca_2FeAlO_5$ and $C_4AF$), whereas the main phase compositions of the sample A3.0 were MgO, f-CaO, $C_3S$, $C_4AF$, and tricalcium aluminate ($Ca_3Al_2O_6$, $C_3A$). This shows that a new $C_3A$ phase was generated and other phases did not change according to the increase in $Al_2O_3$ content in the regenerated samples.

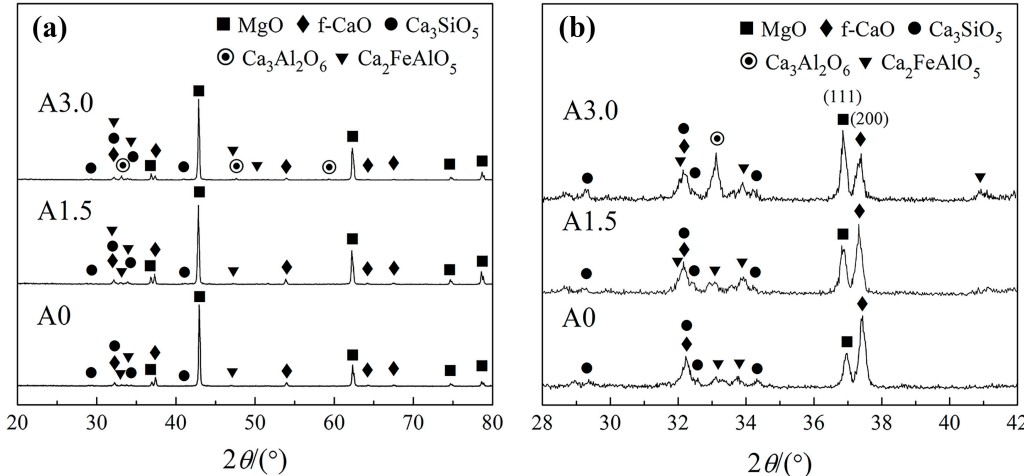

**Figure 3.** XRD patterns of samples A0, A1.5, and A3.0: (**a**) the range of 20–80°; (**b**) the range of 28–42°.

Figure 3b shows that the intensity of the characteristic diffraction peaks of f-CaO [$I_{CaO(200)}$] and MgO [$I_{MgO(111)}$] were the two highest peaks in the range of 28–42° and varied significantly. The relative intensities of $I_{CaO(200)}/I_{MgO(111)}$ decreased observably along with an increase in $Al_2O_3$ content in the regenerated magnesia-calcium bricks, as shown in Figure 4. It also indicates that the content of f-CaO decreased along with an increase in $Al_2O_3$ content in the regenerated samples because the phase of MgO was exclusive and stable in the regenerated samples, as shown in Figure 3.

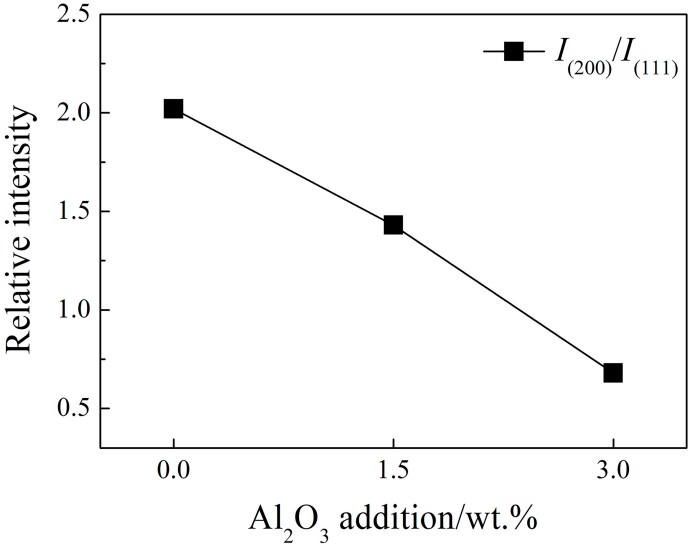

**Figure 4.** Relative intensities of $I_{CaO(200)}/I_{MgO(111)}$ of the samples A0, A1.5, and A3.0.

Figure 3 shows that the element Fe in the regenerated samples existed only in the $C_4AF$ phase, and the element Al in the regenerated samples existed in both the $C_4AF$ and $C_3A$ phases. The standard Gibbs energy of reactions between $Fe_2O_3$, $Al_2O_3$, and f-CaO which generate $C_4AF$ and $C_3A$, are shown in Figure 5 of lines a and c [23]. This shows that the phase of $C_4AF$ generates preferentially when the impurity elements of Fe and Al exist concurrently, and the phase of $C_3A$ generates when there is excess Al element impurity (the mole ratio of $Fe_2O_3$ to $Al_2O_3$ < 1). In addition, a preliminary study proved that the phase of dicalcium ferrite ($Ca_2Fe_2O_5$, $C_2F$) is generated when the mole ratio of $Fe_2O_3$ to $Al_2O_3$ is >1 [20], and the standard Gibbs energy of the reaction between $Fe_2O_3$ and f-CaO for generating $C_2F$ is shown in Figure 5 of line b [23].

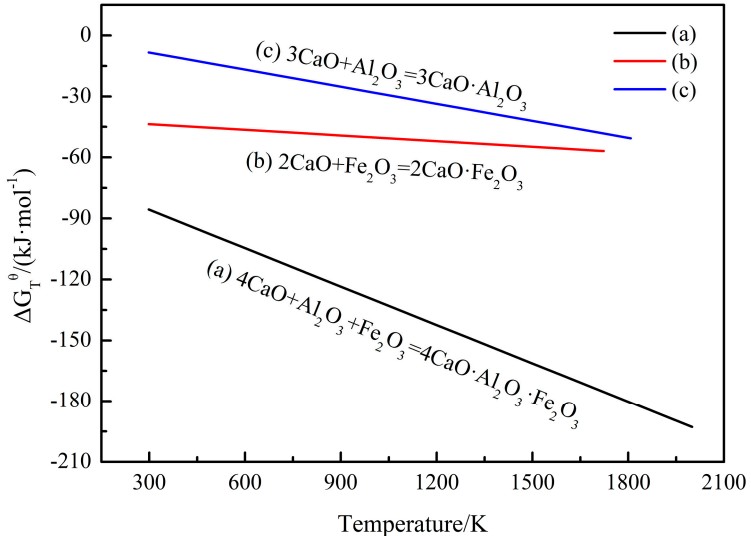

**Figure 5.** Standard Gibbs energy of the reactions between $Fe_2O_3$, $Al_2O_3$, and f-CaO vs. temperature in the regenerated samples.

The theoretical content of the $C_3S$, $C_4AF$ $C_2F$, $C_3A$, and f-CaO phases (calcium-based phases) in the regenerated samples can be calculated using Equations (1)–(7) [24]. The results (as shown in Table 3) indicate that with an increase in $Al_2O_3$ content in the regenerated samples, the form of the element impurity of Fe in the regenerated samples changed from the phase of $C_2F$ to $C_4AF$, whereas the form of the element impurity of Al in the regenerated samples changed from the phase of $C_4AF$ to $C_4AF$ and $C_3A$. The theoretical content of the phase $C_2F$ in sample A0 and the phase $C_3A$ in sample A1.5 were both less than 5 wt.%; therefore, their character diffraction peaks are not observed in Figure 3.

$$w(C_3S) = 3.80w(SiO_2) \tag{1}$$

$$w(C_4AF) = 4.77w(Al_2O_3) \text{ (mole ratio of } Fe_2O_3 \text{ to } Al_2O_3 > 1) \tag{2}$$

$$w(C_4AF) = 3.04w(Fe_2O_3) \text{ (mole ratio of } Fe_2O_3 \text{ to } Al_2O_3 < 1) \tag{3}$$

$$w(C_2F) = 1.70[w(Fe_2O_3) - 1.57w(Al_2O_3)] \tag{4}$$

$$w(C_3A) = 2.65[w(Al_2O_3) - 0.64w(Fe_2O_3)] \tag{5}$$

$$w(\text{f-CaO}) = w(CaO) - 1.10w(Al_2O_3) - 0.70w(Fe_2O_3) - 2.80w(SiO_2) \text{ (mole ratio of } Fe_2O_3 \text{ to } Al_2O_3 > 1) \tag{6}$$

$$w(\text{f-CaO}) = w(\text{CaO}) - 1.65w(\text{Al}_2\text{O}_3) - 0.35w(\text{Fe}_2\text{O}_3) - 2.80w(\text{SiO}_2) \ (\text{mole ratio of Fe}_2\text{O}_3 \text{ to Al}_2\text{O}_3 < 1) \qquad (7)$$

$w(\text{C}_3\text{S})$, $w(\text{C}_4\text{AF})$, $w(\text{C}_2\text{F})$, $w(\text{C}_3\text{A})$, and $w(\text{f-CaO})$—the mass fraction of the calcium-based phases in the regenerated samples, wt.%;

$w(\text{SiO}_2)$, $w(\text{Al}_2\text{O}_3)$, $w(\text{Fe}_2\text{O}_3)$, and $w(\text{CaO})$—the mass fraction of the regenerated samples' composition, wt.%.

**Table 3.** Theoretical contents of each calcium-based phase in the regenerated samples (wt.%).

| Calcium-Based Phases | A0 | A1.5 | A3.0 |
|:---:|:---:|:---:|:---:|
| $\text{C}_3\text{S}$ | 10.11 | 9.98 | 9.85 |
| $\text{C}_4\text{AF}$ | 5.25 | 6.62 | 6.53 |
| $\text{C}_2\text{F}$ | 0.82 | 0.00 | 0.00 |
| $\text{C}_3\text{A}$ | 0.00 | 3.05 | 6.81 |
| f-CaO | 13.11 | 10.66 | 8.12 |

*3.2. Microstructures and Elements Distribution of the Regenerated Magnesia-Calcium Bricks*

Figure 6a,c,e show the SEM images of the regenerated samples' surfaces, and Figure 6b,d,f show the element distribution of the regenerated samples' surfaces. It shows that the white phases shown in the selected areas 1, 2, and 3 in Figure 6a,c,e are f-CaO, $\text{C}_4\text{AF}$ or $\text{C}_4\text{AF-C}_3\text{A}$, and $\text{C}_3\text{S}$ phases, respectively, whereas the dark phases shown in Figure 6a,c,e are all MgO phases based on the element distribution shown in Figure 6b,d,f, and these calcium-based phases became more and more continuous along with an increase in $\text{Al}_2\text{O}_3$ content in the regenerated samples. Moreover, it indicates that the phases of $\text{C}_4\text{AF}$ and $\text{C}_3\text{A}$ showed a tendency of aggregation in samples A1.5 and A3.0. This is because the elements of Fe and Al existed in the form of a liquid phase at the firing temperature (1873 K), and the $\text{C}_3\text{A}$ phase (melting point of 1808 K [23]) and $\text{C}_4\text{AF}$ phase (melting point of 1688 K [25]) successively crystallized from the liquid phases and sintered together during the furnace cooling process.

Figure 7a–c show the SEM images of the regenerated samples' fractures, and Figure 7d shows the bulk density of the regenerated samples. It can be seen that the compactness of the regenerated samples was enhanced along with an increase in $\text{Al}_2\text{O}_3$ content in the regenerated samples. This is because the content of the liquid phase in the regenerated samples at the firing temperature increased along with an increase in $\text{Al}_2\text{O}_3$ content in the regenerated samples, and it improved the sinterability of the regenerated samples.

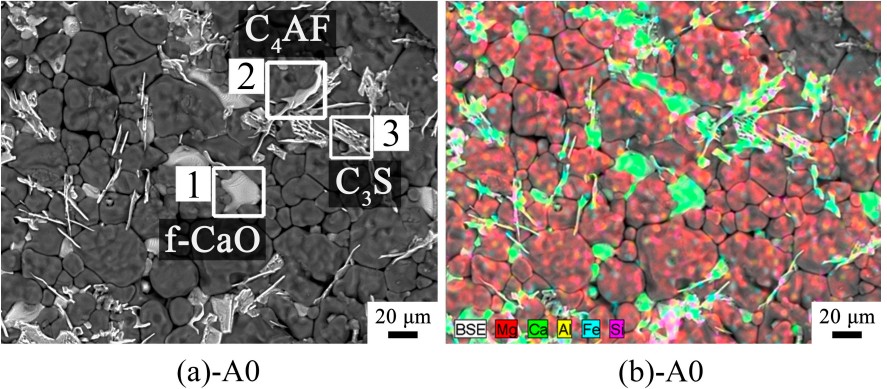

(a)-A0        (b)-A0

**Figure 6.** *Cont.*

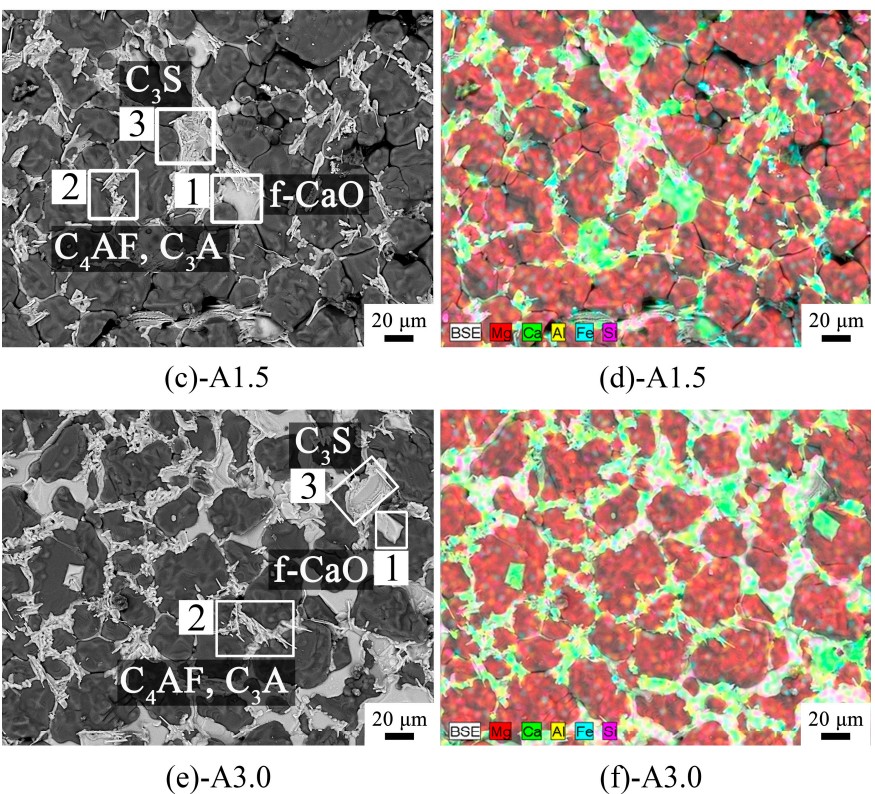

**Figure 6.** (**a**,**c**,**e**) SEM images of the regenerated samples' surfaces; (**b**,**d**,**f**) elements distribution of the regenerated samples' surfaces.

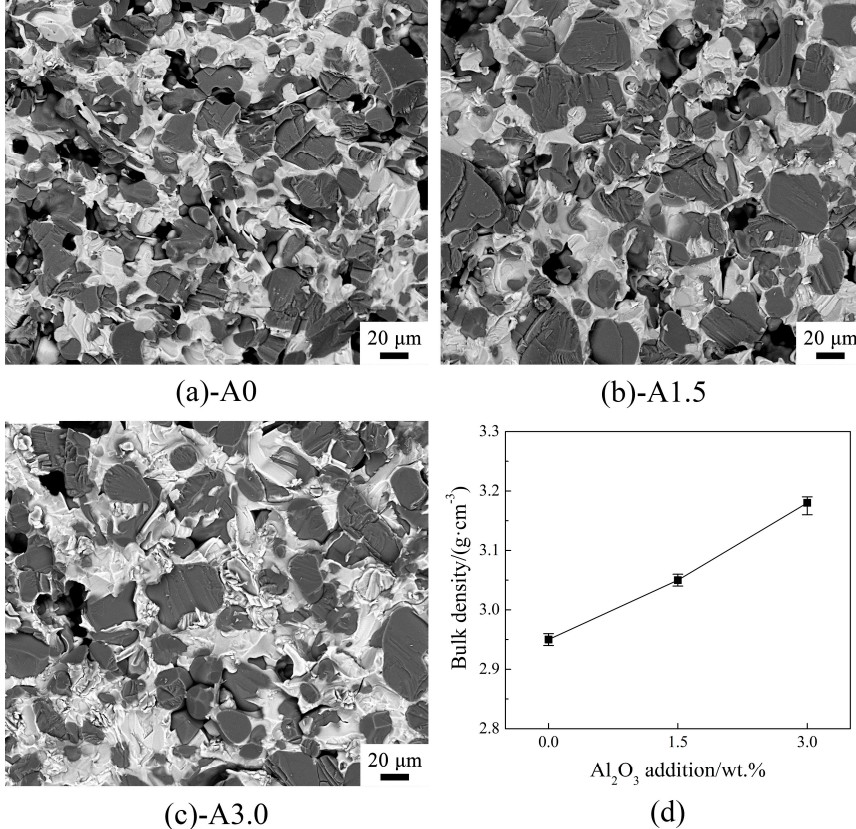

**Figure 7.** (**a**–**c**) SEM images of the regenerated samples' fractures; (**d**) bulk density of the regenerated samples.

### 3.3. Room Temperature and Hot Flexural Strength of the Regenerated Magnesia-Calcium Bricks

Figure 8 shows the results of the room temperature and the hot (1573 K) flexural strength of samples A0, A1.5, and A3.0. This indicates that the room-temperature flexural strength of the regenerated samples increased, and the hot flexural strength of the regenerated samples reduced with $Al_2O_3$ content increasing. This is because, with an increase in $Al_2O_3$ content in the regenerated samples, the density of the regenerated sample increased on the one hand, as shown in Figure 7d, leading to an increase in its room-temperature flexural strength, and on the other hand, the regenerated sample generated more low-melting-point phases (such as $C_4AF$ and $C_3A$), as shown in Table 3, resulting in a decrease in its hot flexural strength.

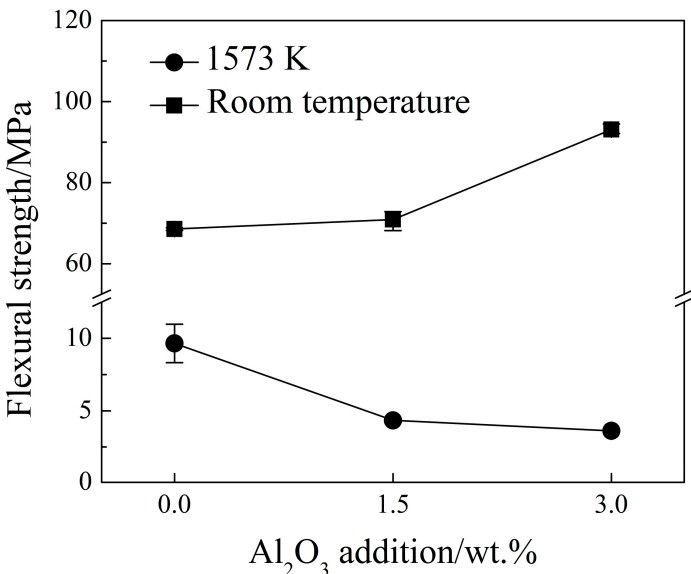

**Figure 8.** Room temperature and hot flexural strength of the samples A0, A1.5, and A3.0.

### 3.4. Kiln Coating Adherence of the Regenerated Magnesia-Calcium Bricks

The flexural strengths of the bonded samples BA0, BA1.5, and BA 3.0 are shown in Figure 9a. It shows that the flexural strengths of the bonded samples were enhanced along with an increase in $Al_2O_3$ content in the regenerated samples; that is, there was an increasing tendency of the kiln coating adherence as $Al_2O_3$ content in the regenerated samples increased.

The SEM images of the cross-section of the bonded samples BA0, BA1.5, and BA3.0 after the kiln coating adherence experiment are shown in Figure 9b–d. It can be seen that the cement clinker corroded the regenerated samples along the calcium-based phases, and the corrosion was more serious in bonded samples BA1.5 and BA3.0 than in bonded sample BA0. This is because the low-melting-point phases in the calcium-based phases of the regenerated samples existed in the form of a liquid phase at the experimental temperature (1823 K), which improved the wettability of the regenerated samples, and thus, the melting cement clinker was much easier to corrode the regenerated samples along the areas of these low-melting-point phases (indicated by red arrows in Figure 9b–d). Therefore, more cement clinker corroded the regenerated sample along with an increase in $Al_2O_3$ content in the regenerated samples as the content of the low-melting-point phases increased, and it also increased the reaction areas between the cement clinker and regenerated samples, which enhanced the kiln coating adherence, as shown in Figure 9a.

Moreover, the reaction between the cement clinker and the phase of MgO, which was the major phase of the regenerated sample, was carried out by an element distribution scanning analysis across their contact areas, as shown in Figure 10a. This indicates that a reaction layer containing Ca, Sim, and Mg existed on the interfacial surface of these contact areas. The thickness *d* of the reaction layer gradually became larger with increasing

Al$_2$O$_3$ content in the regenerated samples, as shown in Figure 10b, which also indicates aggravating corrosion of the cement clinker to the phase of MgO.

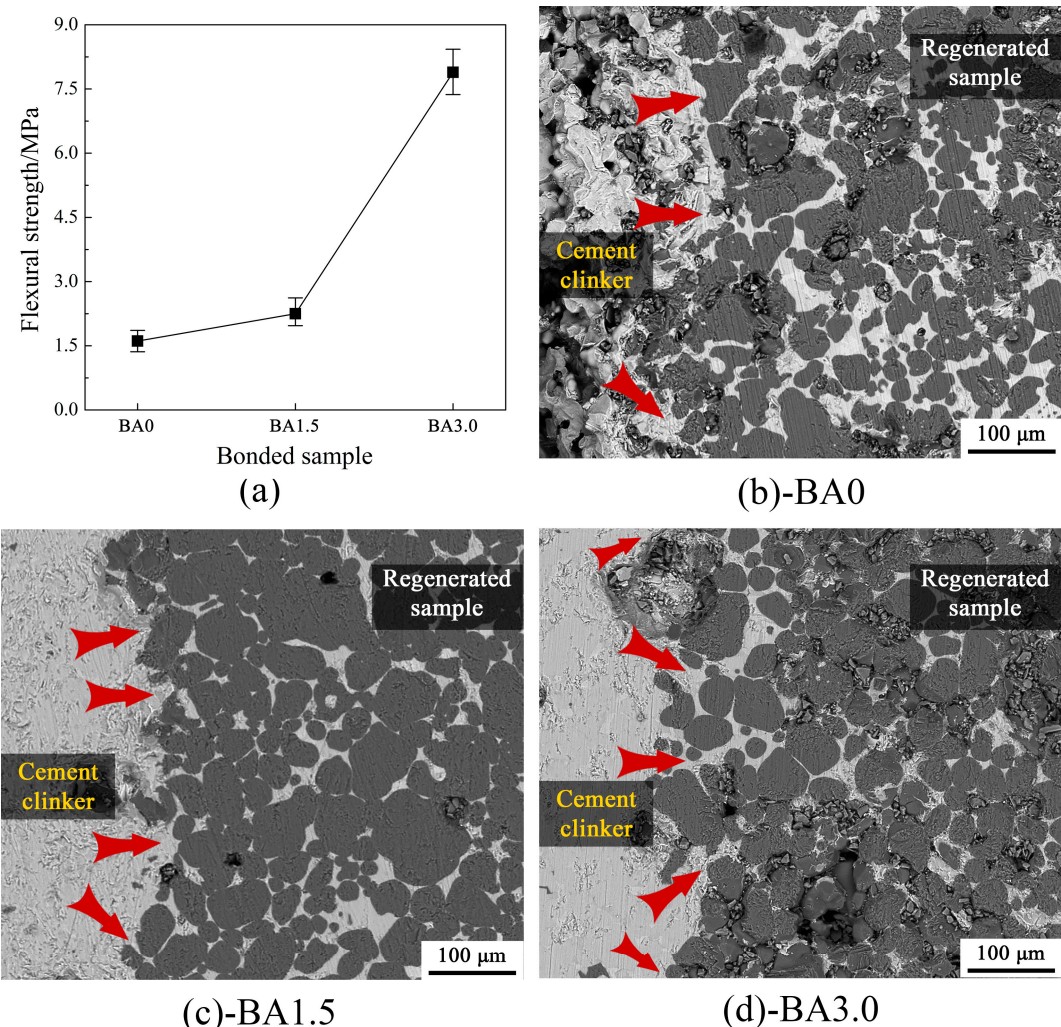

**Figure 9.** (**a**) Flexural strength of the bonded samples; (**b**–**d**) SEM images of the cross-section of the bonded samples after the kiln coating adherence experiment.

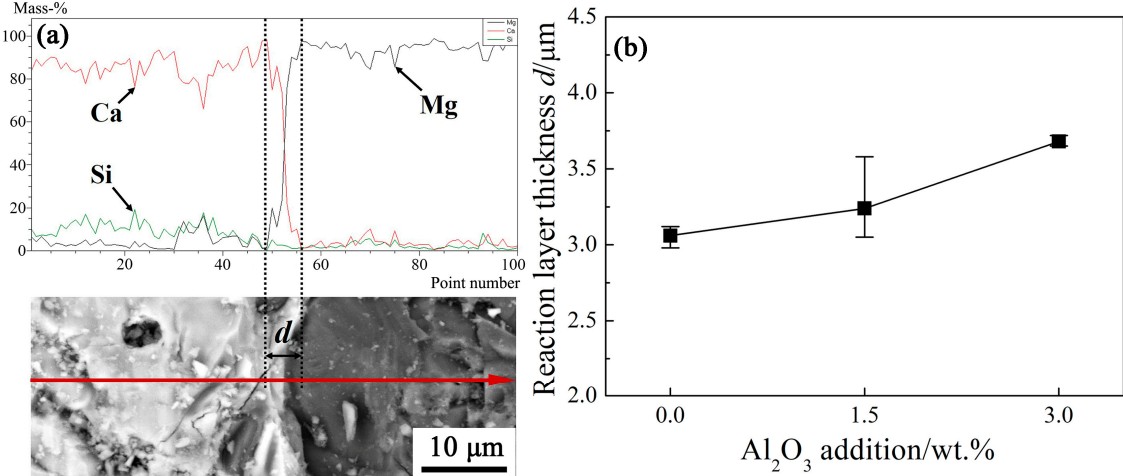

**Figure 10.** (**a**) The element distribution scanning analysis across the contact area of the cement clinker and the MgO phase; (**b**) reaction layer thickness *d* of samples A0, A1.5, and A3.0.

Generally, the increase in $Al_2O_3$ content in the regenerated samples led to an increase in the content of the low-melting-point phases of $C_4AF$ and $C_3A$, and it had a positive effect on the bulk density, room-temperature flexural strength, and kiln coating adherence of the regenerated samples; on the contrary, it had an adverse effect on the hot flexural strength and the corrosion resistance of the cement clinker.

The performances of chromium-free refractories used in the cement rotary kiln as reported in [26,27] were compared with the corresponding performances of the regenerated samples, and the results are shown in Table 4. It shows that the bulk density, room-temperature flexural strength of the samples A0, A1.5, and A3.0, and the flexural strengths of the bonded samples BA0, BA1.5, and BA 3.0 were all higher than those reported in the literature; however, only the hot flexural strength of sample A0 was higher than that reported in the literature. Therefore, sample A0 behaved the best comprehensive performances; that is, the content of $Al_2O_3$ in the regenerated magnesia-calcium brick should not be higher than 1.1 wt.%.

**Table 4.** Comparison of the performances between chromium-free refractories and the regenerated samples.

| Performances | Sample A0/BA0 | Sample A1.5/BA1.5 | Sample A3.0/BA3.0 | Literature |
|---|---|---|---|---|
| Bulk density ($g/cm^3$) | 2.95 | 3.05 | 3.18 | 2.93 |
| Room-temperature flexural strength (MPa) | 68.58 | 70.91 | 93.11 | 21.60 |
| Hot flexural strength (MPa) | 9.65 | 4.34 | 3.62 | 5.40 |
| Bonded sample Room-temperature flexural strength (MPa) | 1.61 (BA0) | 2.25 (BA1.5) | 7.89 (BA3.0) | 0.52 |

## 4. Conclusions

In this work, the effects of $Al_2O_3$ content on regenerated magnesia-calcium bricks for the cement rotary kiln were systematically researched. The regenerated magnesia-calcium brick samples were prepared using spent magnesia-calcium brick, fused magnesia, and $Al_2O_3$ powder additive as the raw materials, liquid paraffin as the binder, and firing at 1873 K for 2 h under an air atmosphere. The phase composition of the regenerated magnesia-calcium bricks with different $Al_2O_3$ contents indicated that the main phases of samples A0 and A1.5 were magnesia (MgO), free calcium oxide (f-CaO), tricalcium silicate ($C_3S$), and tetracalcium aluminoferrite ($C_4AF$). In sample A3.0, a new phase of tricalcium aluminate ($C_3A$) was observed besides MgO, f-CaO, $C_3S$, and $C_4AF$. The increase in $Al_2O_3$ content in the regenerated samples mainly led to an increase in the content of the $C_4AF$ and $C_3A$ phases in the regenerated samples according to the theoretical calculation. The content of $Al_2O_3$ in the regenerated samples has a significant impact on the performance of cement rotary kiln applications. Specifically, as the content of $Al_2O_3$ increased, the content of the low-melting-point phases of $C_4AF$ and $C_3A$ increased, which improved the sinterability of the regenerated sample, leading to an increase in the bulk density, room-temperature flexural strength, and the kiln coating adherence of the regenerated sample. The high-temperature performances of the regenerated samples, such as the hot flexural strength (1573 K) and the corrosion resistance to cement clinker, deteriorated along with an increase in $Al_2O_3$ content because the melting points of the $C_4AF$ and $C_3A$ phases were close to the experimental temperature. Furthermore, $C_4AF$ and $C_3A$ were in the liquid phase form at 1823 K, which wetted the regenerated sample to enhance the reaction between the cement clinker and the regenerated samples, resulting in increasing the kiln coating adherence. Based on the performance and application scenarios of the regenerated

magnesium-calcium brick, the total content of $Al_2O_3$ in the regenerated magnesia-calcium brick should be no more than 1.1 wt.% compared to the main performances with other chromium-free refractories for cement rotary kiln. Moreover, sample A0 can be considered the most suitable regenerated sample for the cement rotary kiln because of its excellent comprehensive performance with a bulk density of 2.95 g/cm$^3$, room temperature and hot flexural strength of 68.58MPa and 9.65 MPa, bonded sample flexural strength of 7.89 MPa, and good corrosion resistance to cement clinker. This may provide a reference for the replacement of hazardous magnesite-chrome bricks in the cement rotary kiln with the regenerated magnesia-calcium bricks.

**Author Contributions:** Conceptualization, G.-B.Q. and M.Z.; methodology, G.-B.Q., Y.-D.H. and M.Z.; software, G.-B.Q. and H.-G.W.; validation, J.H. and X.-H.Z.; formal analysis, G.-B.Q. and J.H.; investigation, G.-B.Q., Y.-D.H. and B.P.; resources, J.H. and B.P.; data curation, G.-B.Q., H.-G.W. and X.-H.Z.; writing—original draft preparation, G.-B.Q.; writing—review and editing, M.Z., Y.-D.H. and H.-G.W.; visualization, G.-B.Q., J.H. and X.-H.Z.; supervision, B.P. and M.Z.; project administration, G.-B.Q. and B.P. All authors have read and agreed to the published version of the manuscript.

**Funding:** This work was supported by the 2023 Zhanjiang Ocean Young Talent Innovation Project of China (2023E0009) and the National Natural Science Foundation Outstanding Youth Science Fund project of China (52322806).

**Institutional Review Board Statement:** Not applicable.

**Informed Consent Statement:** Not applicable.

**Data Availability Statement:** Not applicable.

**Conflicts of Interest:** The authors declare no conflict of interest.

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
