# Peer review of "Effects of Aluminium Oxide Content on the Regenerated Magnesia-Calcium Bricks for Cement Rotary Kiln"

_processes, doi:10.3390/pr11103018_

Round 1

Reviewer 1 Report

An interesting topic and adequate amount of results and discussion. The authors are required to address the following comments

Indicate clearly the novelty of the research and the difference between the current and past research

Improve caption titles

Table 3. Theoretical contents of each calcium–based phases [[in sample A0, A1.5 and A3.0 (wt.%).]]

cordingly. It shows that sample A0, A1.5, A3.0 could all meet the room temperature flexural strength and the strength (which strength???) of coating adherence. However, the hot flexural strength of

ENGLISH..requires improvement e.g in square brackets

There are [[more and more]] spent magnesia-calcium bricks not being reused effectively 54 [15], because of high impurity content such

The investigation and [[comparative]] of their phases, microstructures, room temperature and hot flexural strength, resistance to 68 cement clinker corrosion and strength of coating adherence were carried out [[successively]]

The mixture of the regenerated magnesia-calcium brick green body was prepared by using the main raw materials as show in Table 1, and liquid paraffin as the binder [[firstly]]

In this work, three kinds of regenerated magnesia-calcium brick [[green bodies]] containing 0 wt

NOT TYPICAL… {This section may be divided by subheadings. It should provide a concise and precise description of the experimental results, their interpretation, [[as well as the experimental conclusions that can be drawn.]]}

The theoretical content of the C3S, C4AF C2F, C3A and f–CaO phases (calcium–based phases) in the regenerated samples can be calculated by the Eq. (1) - (7) [21]. The results (as shown in Table 144 3) [[indicates]] that the theoretical content of the C3A phase increased while that of the f–CaO phase decreased as the increasing of Al2O3 addition, and the Fe element of the sample A1.5 and A3.0 all generated the C4AF phase. These [[calculation]] [[results]] were identical with the phase compositions analysis shown in Figure 3.

samples were composed of the C3S, C4AF, C3A and f–CaO phases, among which the 206 phases of C4AF and C3A content increased with Al2O3 [[addition increasing]]. It improved

The [[analysis results]] indicate that a reaction layer containing element Ca, Si and Mg existed 221 in the interfacial surface of these contact areas. The

There was a solid solution appearance between the Al element and the C3S phase, resulting in the actual content of the C3A phase [[was lower]] than the theoretical calculation, and it led to the phase

An interesting topic and adequate amount of results and discussion. The authors are required to address the following comments

Indicate clearly the novelty of the research and the difference between the current and past research

Improve caption titles

Table 3. Theoretical contents of each calcium–based phases [[in sample A0, A1.5 and A3.0 (wt.%).]]

cordingly. It shows that sample A0, A1.5, A3.0 could all meet the room temperature flexural strength and the strength (which strength???) of coating adherence. However, the hot flexural strength of

ENGLISH..requires improvement e.g in square brackets

There are [[more and more]] spent magnesia-calcium bricks not being reused effectively 54 [15], because of high impurity content such

The investigation and [[comparative]] of their phases, microstructures, room temperature and hot flexural strength, resistance to 68 cement clinker corrosion and strength of coating adherence were carried out [[successively]]

The mixture of the regenerated magnesia-calcium brick green body was prepared by using the main raw materials as show in Table 1, and liquid paraffin as the binder [[firstly]]

In this work, three kinds of regenerated magnesia-calcium brick [[green bodies]] containing 0 wt

NOT TYPICAL… {This section may be divided by subheadings. It should provide a concise and precise description of the experimental results, their interpretation, [[as well as the experimental conclusions that can be drawn.]]}

The theoretical content of the C3S, C4AF C2F, C3A and f–CaO phases (calcium–based phases) in the regenerated samples can be calculated by the Eq. (1) - (7) [21]. The results (as shown in Table 144 3) [[indicates]] that the theoretical content of the C3A phase increased while that of the f–CaO phase decreased as the increasing of Al2O3 addition, and the Fe element of the sample A1.5 and A3.0 all generated the C4AF phase. These [[calculation]] [[results]] were identical with the phase compositions analysis shown in Figure 3.

samples were composed of the C3S, C4AF, C3A and f–CaO phases, among which the 206 phases of C4AF and C3A content increased with Al2O3 [[addition increasing]]. It improved

The [[analysis results]] indicate that a reaction layer containing element Ca, Si and Mg existed 221 in the interfacial surface of these contact areas. The

There was a solid solution appearance between the Al element and the C3S phase, resulting in the actual content of the C3A phase [[was lower]] than the theoretical calculation, and it led to the phase

Reviewer 2 Report

The article is interesting and original, but needs improvement.

1. Introduction

- Authors provide good and relevant information in the introduction, however, they do not clearly state the purpose of the research. The information at the end of the introduction on lines 66-69 should be rewritten. In the least interpretation, authors simply list the research methods and objects of the investigation. I recommend to the authors to clearly state the purpose of the research instead of this information. 

2. Materials and Methods

- It is necessary to provide a reference to the methodology described in subsection 2.2 on lines 82-87.

- It is not clear why exactly 4 additive concentrations were investigated. What happens if the additive concentration is 4%? Recommend to the authors to give an explanation about the amount of additives. Perhaps the authors can refer to some previous researches.

- The methodological part of subsection 2.3 should be expanded. Recommend to the authors to prescribe the modes of research. At the moment, the authors recite the devices without going into details. 

- The authors investigated the corrosive destruction of materials. However, in the methodological part the authors do not mention the conditions of corrosion tests (medium, temperature, etc.). The authors should provide appropriate explanations.

3. Results

- I recommend the authors to be more careful. Information from the template of this journal should be removed from lines 111-113.

- The results of the adhesion and corrosion studies are described very vaguely. Regarding the corrosion studies, it is not clear how the authors estimate corrosion. There are no original samples available. Recommend that the authors expand the description of the results of the adhesion and corrosion studies. At a minimum, photos of original specimens should be added to make the results more visual.

4. Conclusions

- The conclusions strongly overlap with the abstract. It is requested that the authors rewrite the conclusions. It should be stated more clearly and locally.

5. Figures

- Figure 6: a master scale marker should be indicated for enlarged sections.

Round 2

Reviewer 1 Report

The authors have largely addressed my comments. Some editing still required e.g.

is Equation 1 necessary (well known equation)

section 3.3. English requires some  improvement (e.g. line 261.oppositely,. also the last long sentence)

The authors have largely addressed my comments. Some editing still required e.g.

is Equation 1 necessary (well known equation)

section 3.3. English requires some  improvement (e.g. line 261.oppositely,. also the last long sentence)

Reviewer 2 Report

Thanks to the authors for a job well done. After the changes made, the material of the article has improved considerably. I recommend the authors to check the text for merged words, for example, lines 84,87,88.

Reviewer 3 Report

Please find the attachment
